# Simple mechanistic traits outperform complex syndromes in predicting avian dispersal distances
Guillermo Fandos [1,2] ✉, Robert A. Robinson [3,4] & Damaris Zurell [1]

Dispersal is a fundamental ecological and evolutionary process, but identifying its determinants and predicting it across species remains a major challenge. Dispersal syndromes, which describe patterns of covariation among traits related to dispersal, are thought to capture general rules of dispersal evolution and its ecological consequences. Based on the most comprehensive empirical dispersal dataset available for European birds, we test how dispersal syndromes form and how well they predict dispersal across species. We found that distinct dispersal processes were governed by different trait combinations, with body mass consistently predicting overall dispersal, whereas flight efficiency was key for long-distance dispersal events. However, multi-trait dispersal syndromes performed poorly for phylogenetically distant species and were outperformed by models based on single mechanistic traits, especially body mass, life history, and, to a lesser extent, flight efficiency. Thus, single traits with clear mechanistic meaning predict avian dispersal ability better than complex syndromes. These findings highlight the complexity of avian dispersal and emphasize the need for refined mechanistic approaches to understand the constraints shaping dispersal evolution. Together, our study calls for broader empirical efforts and more mechanistic frameworks to uncover the evolutionary and ecological drivers of dispersal.

Dispersal —the movement from a natal or breeding site to another breeding site— is a key ecological process that determines connectivity between populations and, by that, the population dynamics, gene flow, range dynamics, and scope for local adaptations[1–3]. Understanding dispersal is essential for predicting how species respond to global change. Yet, dispersal remains one of the least understood ecological processes because it emerges from complex trade-offs between selective pressures, environmental context, and covariation among traits[4,5]. Dispersal syndromes reflect these patterns of covariation between morphological, physiological, or behavioral traits and dispersal[6], offering critical insights into the evolution and causes of dispersal. Empirically measured dispersal data are scarce[7,8], so dispersal syndromes have only been studied for a limited number of species[9], and generally as meta-analyses across narrow taxonomic groups. At the same time, dispersal information is crucial for predicting the species' spatial dynamics under climate and land-use change[3,10,11], often requiring extrapolation for species lacking empirical data. Thus, a deeper understanding of dispersal traits and dispersal syndromes is also key to improving our predictive capacity and to

evaluating the generality of trait-based approaches to anticipate future global change impacts on biodiversity.

Birds are one of the best-studied taxa, and indeed several studies have attempted to understand the covariation of dispersal[12–15] or dispersal proxies[16] with species traits. From theoretical and empirical studies, we can expect that dispersal syndromes are multidimensional and that dispersal correlates with morphological, behavioral and life-history traits[6]. In birds, the cost of movement plays a predominant role in shaping dispersal patterns, with variation in dispersal distance often correlating with flight-efficiency proxies such as wing aspect ratio[15]. While body size is theoretically linked to dispersal in active dispersers[17] and widely used as a proxy for dispersal ability, recent analyses suggest its role may be more limited[13,15]. Beyond morphology, behavior and physiology can strongly influence metabolism and, hence, the locomotor activity of a species[18,19]. Functional and behavioral traits such as feeding guild[20], migratory behavior[12,21], or life-history traits (e.g., survival, age at maturity, and fecundity[22]) may also influence dispersal, for example, through variation in the behavioral strategies, spatial requirements, or

[1]Institute for Biochemistry and Biology, University of Potsdam, Potsdam, Germany. [2]Department of Biodiversity Ecology and Evolution, Faculty of Biology, Complutense University, Madrid, Spain. [3]British Trust for Ornithology, The Nunnery, Thetford, Norfolk, UK. [4]European Union for Bird Ringing, Zorroagagaina 11, E20014 San Sebastián, Spain. ✉e-mail: gfandos@ucm.es; fandos-guzman@uni-potsdam.de

kin-competition avoidance mechanisms. However, these same traits were shown to have a positive, negative, or no relationship to dispersal, depending on the study[12,15,23]. Hitherto, most relationships between dispersal and morphological, behavioral, and functional traits seem idiosyncratic among bird species (except perhaps for flight efficiency[13,15]) and lack robust empirical support, making identifying dispersal syndromes difficult[24].

Progress has also been hindered by methodological limitations. Most previous studies used inconsistent definitions of dispersal, used raw dispersal distances that can be skewed by extreme dispersal events, or lacked the data needed to detect these long-distance dispersal events[9,25]. Dispersal kernels, which describe the full probability distribution of dispersal distances, offer a more mechanistic representation of dispersal movements and allow comparison across species and taxa[12,13,15]. Yet, standardized empirical dispersal kernels are rare. Identifying the set of traits that shape short and long-distance dispersal syndromes is important to fully understand the mechanistic determinants of dispersal, the constraints associated with movement, and, more generally, the trade-offs underlying dispersal evolution.

An open question is whether dispersal syndromes provide general and predictable insights into the dispersal process, enabling inference for species lacking data[26]. Previously, simplified or theoretically derived dispersal syndromes have been used for this purpose[27,28], as empirical dispersal estimates are scarce[7,8,12]. These studies have used species-level traits (body mass, wingspan or wing morphology, migratory behavior, life history[27–29]) and phylogenetic relatedness[10] to predict dispersal for diverse taxonomic groups. Yet, the predictive potential and generality of these trait-based approaches remain limited by conceptual and methodological constraints. One limitation arises from the potential confusion between dispersal (the movement between consecutive breeding sites) and other types of geographic movements (including nomadic or seasonal movements). Birds rely on highly adaptive traits to undertake seasonal migrations[30,31] that may not be identical to the traits relevant for natal and breeding dispersal[21,32] as the dispersal process involves not only movement between sites but also complex emigration and settlement decisions, and the evolutionary causes and fitness benefits of dispersal and migration differ. Furthermore, as scarce empirical dispersal data have impeded the robust assessment of these dispersal inferences across many species, families, and orders, little is known about how generalizable or predictable dispersal syndromes are between closely to distantly related species. Recent open-access databases on bird dispersal[8], phylogeny[33], and species traits[34,35] now provide an unparalleled opportunity to address these questions using standardized data across a wide taxonomic range.

Here, building on standardized empirical dispersal distance kernels from ref. 8, we investigate the formation and the predictive capacity of dispersal syndromes in European bird species. Birds are ecologically diverse and show a range and variability of traits across species, which allows robust comparisons across the group and facilitates comprehensive analyses of dispersal syndromes. Also, birds are the only group for which standardized dispersal kernels exist across many closely and distantly related species. We first investigate systematic covariations between (natal and breeding) dispersal ability and a suite of traits while controlling for how common ancestry influences this covariation. We include traits associated with demography, morphology, ecological specialization, and behaviors relevant to movement[36] and assess their overall importance for explaining interspecific variation in dispersal. Within this analysis, we also test for consistency of dispersal syndromes between median-distance versus long-distance dispersal (extracted from the dispersal kernels[8]). Finally, we apply a cross-validation statistical approach to determine how single-trait and multi-trait models can predict dispersal distances, both within and across orders. This combined approach allows us to assess not only which traits shape dispersal but also how well single mechanistic traits and multi-trait syndromes can predict dispersal in species lacking empirical data, providing valuable insights into the generality and limits of trait-based inference in dispersal ecology.

## Results
### Assessing dispersal syndromes

Median and long-distance dispersal estimates were available for 234 species, and separate natal and breeding dispersal estimates for 113 European breeding bird species[8]. Using two heavy-tailed kernels that best captured dispersal in European birds[8], we fitted a multi-trait phylogenetic regression model to test covariations between bird dispersal ability (median and long-distance) and a suite of relevant morphological, ecological and biogeographic traits (Supplementary Table 1): body size, Hand Wing Index (HWI), slow-fast continuum of life histories (from long-lived species with high adult survival representing the slower end to early maturing species with short generation times and high reproductive rates at the fast end[37]), habitat openness, feeding guild, breeding latitude, and migration tendency. We removed species with missing values for any trait (Supplementary information 1), and the complete datasets with all traits included 138 species for total dispersal, 63 for breeding dispersal, and 72 for natal dispersal. Dispersal syndromes derived from the Weibull and Half-Cauchy distributions were consistent (Fig. 1, Supplementary Fig. 7); therefore, we present results using the Weibull distribution for simplicity. While we also assessed dispersal syndromes for natal and breeding dispersal, here we focus on total dispersal (including breeding and natal dispersal events), with detailed results for natal and breeding dispersal, including univariate trait relationships and species-level patterns, provided in Supplementary Note 3 (Supplementary Figs. 4–7).

We first assessed model fit and variable importance using projection predictive inference, which evaluates how well different sets of traits explain observed dispersal distances within the dataset. Variable-selected models consistently outperformed both univariate and full models for median dispersal (Supplementary Table 2). For total dispersal, the variable-selected model identified body mass ($z = 0.504$), life history ($z = -0.383$), and latitude ($z = -0.193$) as the strongest predictors of median dispersal distances (Fig. 1: Supplementary Table 3) and achieved strong predictive performance ($R^2 = 0.522$ marginal, $0.668$ conditional; Supplementary Table 3). Body mass showed a consistent positive effect, indicating that larger species disperse farther (Fig. 2). In contrast, life history had a negative effect, with faster species (short-lived, highly reproductive) dispersing more (Fig. 2). Dispersal decreased with latitude and showed a weaker negative association with habitat openness (Fig. 1: Supplementary Table 4).

Long-distance dispersal showed markedly different patterns, being best explained by flight efficiency alone. Hand Wing Index (HWI) was the single best predictor ($z = 0.335$), but with limited explanatory power ($R^2 = 0.034$ marginal: Supplementary Table 3), though model averaging approaches combining HWI and diet substantially improved predictive performance ($R^2 = 0.513$). This emphasizes the distinct mechanisms underlying rare long-distance dispersal events compared to typical median distances. We found strong phylogenetic signals across models for total dispersal ($\lambda = 0.548–0.895$), consistent with substantial phylogenetic structure in dispersal patterns (Supplementary Table 3). However, this pattern may partly reflect correlated ecological strategies within lineages rather than strict evolutionary constraints. Overall, we found similar patterns and robust results when we used only passerines (Supplementary Fig. 9), the largest avian order, which also shares relatively similar life-history traits. This sensitivity analysis confirmed that the trait syndromes identified by the model selection approach were consistent within a more phylogenetically and ecologically homogeneous group. We also explored a limited set of ecologically meaningful interactions (e.g., body mass × life history, body mass × diet), which slightly improved model fit in some cases (Supplementary Note 4). However, given sample size constraints and to ensure consistent model structure across analyses, we focus here on main effects. The interaction models are shown in the Supplementary Note 4 for transparency.

Examining natal and breeding dispersal separately revealed distinct trait associations, indicating that the drivers of dispersal can differ between life stages. Body mass was the strongest predictor for both breeding and natal dispersal ($z = 0.469$ and $z = 0.652$ for median dispersal, respectively;

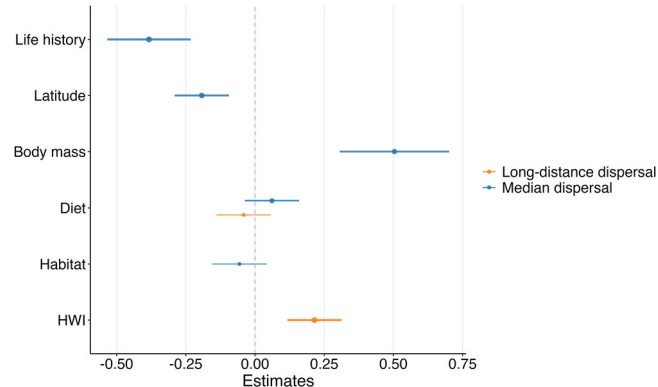

**Fig. 1 | Dispersal syndromes in European birds.** Standardized coefficients and 95% credible intervals of predictors of median (orange circles) and long-distance (blue circles) dispersal among European birds (*n* = 138 species) based on phylogenetic generalized linear mixed models. For median dispersal, coefficients derive from reduced multivariate models with variables retained through projection predictive selection based on expected log predictive density. For long-distance dispersal, coefficients represent Bayesian model-averaged estimates from stacking two competitive univariate models (Hand-Wing Index and diet), weighted by their leave-one-out cross-validation performance. Variable importance is indicated by point size: for median dispersal based on variable selection ranking, and for long-distance dispersal based on stacking weights. Error bars represent 95% credible intervals from posterior distributions. Dispersal estimates stem from the Weibull distribution[8].

$z = 1.027$ and $z = 1.131$ for long-distance dispersal), with moderate to good model performance (breeding: marginal $R^2 = 0.476$ for median and 0.479 for long-distance dispersal; natal: $R^2 = 0.506$ and 0.570, respectively; Supplementary Table 3). The feeding guild showed significant associations only with natal dispersal ($z = 0.402$ for median and $z = 0.612$ for long-distance; Supplementary Table 4) but had no significant effect on breeding dispersal. Breeding dispersal models retained only body mass as a significant predictor, while natal dispersal models included both body mass and feeding guild. Long-distance dispersal showed similar trait associations to median dispersal within each life stage but with consistently stronger effect sizes. For natal long-distance dispersal, body mass showed the strongest effect ($z = 1.131$), followed by feeding guild ($z = 0.612$). For breeding long-distance dispersal, only body mass remained significant ($z = 1.027$). Phylogenetic signals were low for both breeding ($\lambda = 0.139$) and natal ($\lambda = 0.272$) dispersal (Supplementary Table 3), indicating limited phylogenetic structure compared with the stronger values observed for total dispersal models, which may partly reflect correlated ecological strategies within lineages rather than strict evolutionary constraints.

### Predicting dispersal within and across orders

In the next step, we used a cross-validation framework to test how well these models predict dispersal distances across species and taxonomic groups, thereby assessing out-of-sample predictive performance. We assessed the predictive accuracy of single- and multi-trait models to estimate dispersal for missing species within and between orders. For the within-order cross-validation, we used a five-fold design refitting the trait models to 80% of the species and cross-predicting to the hold-out 20% of species. When considering total dispersal, single-trait models, including body mass ($R^2 = 0.454$) and life history ($R^2 = 0.408$), achieved the highest within-order predictive power for the median dispersal. In comparison, the multi-trait model (i.e., the multi-trait dispersal syndrome) only achieved predictive power of $R^2 = 0.263$ for the median dispersal (Fig. 3; Supplementary Table 5). Thus, single-trait models achieved higher within-order predictive performance than the multi-trait model and models calibrated with only phylogeny or randomly (Supplementary Fig. 15). Similar results were found for breeding and natal dispersal, with a consistently high predictive accuracy of body mass. However, for natal dispersal, single-trait models of HWI, habitat openness, and diet also achieved higher within-order predictive

accuracy than the multi-trait model, and for breeding dispersal, only body mass and habitat openness had a higher predictive accuracy than the multi-trait syndrome. Overall, the highest predictive accuracy was achieved for breeding dispersal ($R^2 = 0.638$ for the single-trait model of body mass). For long-distance dispersal, the single-trait model for migration distance had considerably higher within-order predictive accuracy but otherwise the predictive performances were very similar to trait models predicting median dispersal (Supplementary Fig. 14; Supplementary Table 6).

For the between-order cross-validation, we used a four-fold design and selected four bird orders with a reasonable number of species (11–68 species within each order), refitting the trait models with one order and cross-predicted to the other three orders to assess prediction accuracy (Fig. 3; Supplementary Table 6). This is a very conservative scenario reflecting that typically, we have dispersal estimates for fewer species than we predict[10]. Again, for total dispersal, the highest between-order predictive performances were achieved in single-trait models for body mass ($R^2 = 0.392$) and life history ($R^2 = 0.465$), and these models clearly outperformed predictions based only on phylogeny or calibrated randomly (Supplementary Fig. 15). As expected, between-order predictions had lower predictive power than within-order predictions, especially for breeding and natal dispersal (Fig. 3). Similarly, single-trait models of body mass, life history, and also HWI were most informative for between-order predictions of natal and breeding dispersal (Supplementary Fig. 13) Interestingly, within-order predictions and between-order predictive performance were lower for total dispersal than for natal and breeding dispersal, indicating slightly different mechanisms behind these dispersal processes (Fig. 3; Supplementary Fig. 13). When weighting $R^2$ values by the number of species per order, the overall mean predictive performance between orders increased slightly, reflecting that larger groups contributed more stable estimates. Nevertheless, within-order predictions consistently outperformed between-order ones (Supplementary Fig. 16), indicating that model accuracy declines with increasing phylogenetic distance and highlighting the challenge of predicting dispersal across distantly related taxa.

### Discussion

Our comprehensive analysis of dispersal syndromes in European birds revealed consistent covariation between dispersal distances and a suite of morphological, life-history, and ecological traits. Multi-trait models, therefore, provided some additional explanatory power, particularly for median total dispersal, helping to clarify how multiple traits jointly shape movement capacity. However, their advantage was limited and did not translate into stronger predictive performance across species. When we evaluated predictive performance using cross-validation, single-trait models related to body mass, life history, and flight efficiency provided more robust and generalizable predictions, particularly across orders. These traits have clear mechanistic links to dispersal ability and avoid the confounding interactions that can limit the predictive use of multi-trait models. Our work provides unique insights into the determinants of broad-scale interspecific variation in dispersal ability and has important implications for predicting dispersal in species lacking empirical estimates[4,24,38]. Together, these findings highlight that while combining traits can occasionally improve understanding of specific dispersal processes, single mechanistic traits provide a more reliable basis for predicting dispersal across broader phylogenetic scales.

Dispersal syndromes serve as valuable tools in studying and understanding the diverse patterns of dispersal observed among different species. Our goodness-of-fit analyses revealed that multiple selection pressures contribute to determining species dispersal ability, with consistent covariation between dispersal distances and a suite of morphological, life-history, and ecological traits. Body mass emerged as the most consistent and strongest predictor across all dispersal types, supporting theoretical predictions and empirical work that suggest larger species have greater dispersal ability[12,23,24,28,38,39]. This can be attributed to factors such as the size-dependent nature of competitive ability, energy availability, energetic requirements, and costs, as well as how dispersal reduces competition and

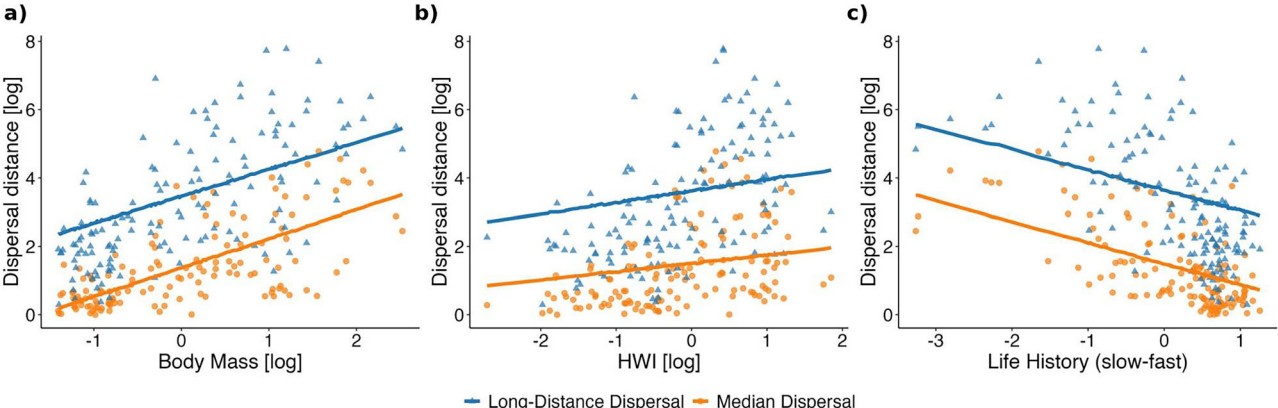

**Fig. 2 | Effect of morphological, ecological, and life history traits on dispersal.** Panels show how median (orange circles) and long-distance (blue circles) dispersal in European birds (*n* = 138 species) varies with **a** body mass (log-transformed), **b** Hand-Wing Index (HWI, log-transformed), and **c** life-history pace (slow–fast continuum). Lines correspond to univariate phylogenetic generalized linear mixed models; solid lines indicate slopes significantly different from zero, dashed lines indicate non-significant relationships. Error bars represent 95% credible intervals. Results from univariate models are provided in Supplementary Table 7. Dispersal estimates stem from the Weibull distribution[8].

avoids inbreeding[13–15,23,40], reflecting the multicausal nature of dispersal decisions[41]. Additional analyses (Supplementary Note 4) indicate that body mass also interacts with other ecological and life-history traits, consistent with theoretical expectations that body size modulates the influence of these traits on dispersal outcomes. These patterns suggest that trade-offs between energy use, competitive ability, and life-history strategy vary with bird size.

Feeding guild showed specific importance for natal dispersal, with carnivorous birds exhibiting higher dispersal tendencies than other trophic groups. This pattern may relate to territoriality and competition dynamics rather than resource availability alone, particularly among birds of prey, where strong territorial behavior and intraspecific aggression can promote greater natal dispersal distances[20,23,40]. These differences between life stages underline that the causes of dispersal are context-dependent and shaped by both social and ecological constraints. The absence of dietary effects in breeding dispersal suggests that once established, adults tend to remain site-faithful if territory quality remains stable. Other ecological traits, such as migratory distance, latitude, and habitat affinity, showed only weak associations with dispersal distances. Nevertheless, high-latitude species tended to show reduced dispersal, consistent with life-history theory predicting stronger selection for time- and energy-constrained reproductive strategies in extreme environments, often prioritizing survival and reproduction over dispersal[42,43]. This supports the idea that dispersal decisions are multicausal and context-dependent, reflecting trade-offs between energetic costs, competitive interactions, and life-history strategies[44].

Hand-wing index (HWI) and wing morphology have been proposed as strong predictors of natal dispersal[13–15,45,46]. Theoretical models of dispersal predict that traits affecting movement capacity (e.g., flight efficiency) primarily influence the transfer phase, whereas other traits (e.g., body size, life history) determine departure and settlement decisions. Framing dispersal within these three phases (departure, transfer, and settlement)[44] helps clarify why proxies of flight efficiency do not consistently predict all aspects of dispersal. In our study, HWI covaried only with total long-distance dispersal, and had a minimal or negligible influence on median, natal, or breeding dispersal. This pattern underscores the difficulty of disentangling movement capacity from dispersal behavior[21,32,47–49] and highlights that proxies of morphology capture only part of the behavioral complexity of dispersal. HWI remains a valuable proxy for flight efficiency, but more precise morphological metrics, such as wing aspect ratio, may better capture the influence of flight capacity on dispersal distances[13,15]. In addition, our within and between-order cross-validation of trait models showed that although flight efficiency had little apparent effect on the dispersal syndromes, it could moderately predict the breeding and natal dispersal distance of other bird species (Fig. 3). Continued collection of standardized

dispersal kernels will be essential to confirm these relationships and to clarify how movement capacity interacts with migration and dispersal decisions in avian populations[32].

Measuring bird dispersal effectively usually requires extensive mark-recapture studies or direct tracking of individuals over multiple years, which are costly and often impractical[9]. Our results show that a trait-based modeling approach, in which models are fitted to large-scale ring recoveries, can be used to predict median and long-distance dispersal distances for species lacking empirical data, while also highlighting the challenges and limitations of such predictions[6,27]. Considering phylogenetic relatedness is certainly important when predicting ecological traits and behaviors[50]. The shared evolutionary history within taxonomic groups facilitates understanding trait covariation and improves predictions within these groups. However, predicting dispersal across taxonomically distant orders with a multi-trait model is challenging due to the wide trait variation, introducing complexity and uncertainty with very limited predictive power at low sample sizes[51]. Although total dispersal is conceptually related to both natal and breeding movements, the empirical kernels used here are derived independently from movement data integrating multiple dispersal events rather than a simple sum of their components. Consequently, differences between total and life-stage-specific dispersal models likely reflect distinct data sources and underlying processes. The strong phylogenetic signal in total dispersal ($\lambda$ = 0.548–0.895) versus weak evolutionary constraints on natal and breeding movements ($\lambda$ = 0.139–0.272) may explain these different cross-taxonomic predictive challenges. Part of this phylogenetic structure likely reflects correlated ecological strategies within lineages rather than strict evolutionary constraints. In contrast, predictions based on single traits with clear mechanistic links to dispersal, such as body size and life history, and to some extent flight efficiency[12,23,27], are more robust when predicting both within and across orders (while controlling for phylogenetic relatedness). Other traits, such as latitude or habitat, or multi-trait syndromes, perform well within closely related species but poorly across orders due to greater trait variability between species across the phylogenetic tree[52,53]. Overall, these patterns emphasize the complexity of dispersal and the different selective pressures and phylogenetic pathways among clades, highlighting that predictive accuracy declines as ecological and evolutionary contexts diverge[54]. Regardless of the underlying mechanisms, the lower predictive performance observed across orders for natal and long-distance dispersal suggests that these movements are more context-dependent, likely because individuals lack prior knowledge about resource availability and mating opportunities in unfamiliar environments[40,55].

Although our study draws on one of the largest standardized dispersal datasets currently available for birds, several important limitations and open

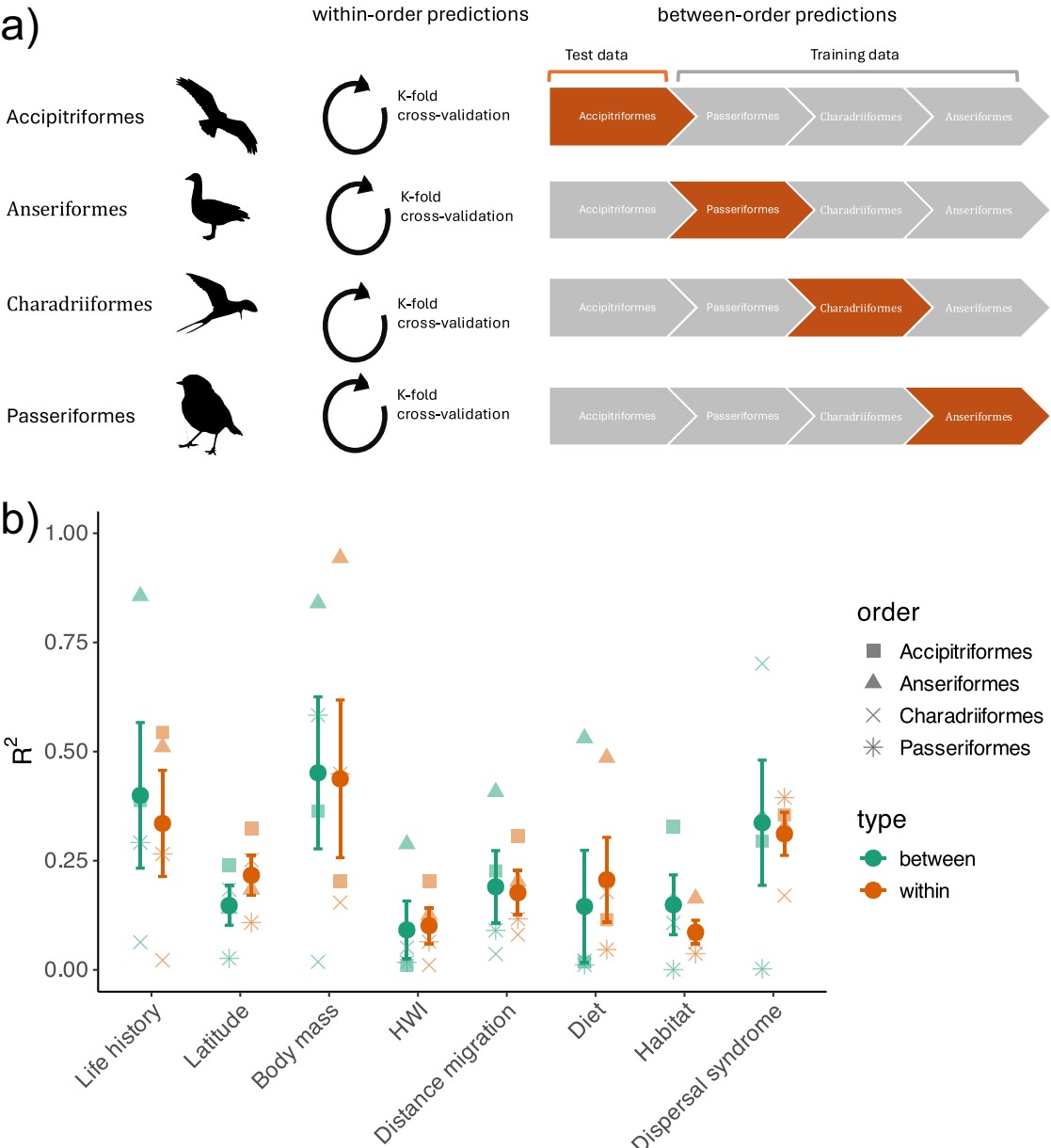

**Fig. 3 | Predictive accuracy of single-trait and multi-trait dispersal models.**
**a** Conceptual overview of cross-validation approaches. Within-order validation: 5-fold cross-validation where each order was split into training (80%) and test (20%) subsets. Between-order validation: each of the four orders was iteratively used as training data to predict dispersal in the remaining three orders. Only orders with ≥11 species were included (Accipitriformes, $n = 11$; Anseriformes, $n = 12$; Charadriiformes, $n = 11$; Passeriformes, $n = 68$). **b** Predictive performance ($R^2$) for single-trait and multi-trait models. Point shapes indicate the order used as training data for between-order predictions (squares: Accipitriformes; triangles: Anseriformes; crosses: Charadriiformes; asterisks: Passeriformes). Colors distinguish within-order (orange) from between-order (teal) validation results. Each point represents the mean $R^2$ across all test predictions for that training order. Error bars represent ±1 standard deviation. Silhouettes from PhyloPic (www.phylopic.org), by T. Michael Keesey, CC0 1.0 Public Domain.

questions remain. First, our analyses were limited to European birds, which may restrict the generalizability of our results to other (especially non-temperate) biomes and taxa with different ecological and evolutionary dynamics. Distinct dispersal syndromes likely emerge in other biogeographic regions under different ecological constraints, seasonal patterns, and landscape structures, particularly in tropical and southern-hemisphere systems. This underscores the need for more geographically comprehensive and standardized global datasets[8]. Second, dispersal kernels are inherently phenomenological and context-dependent, emerging from different underlying mechanisms and drivers[7,56]. Dispersal is a complex process, and the three phases of dispersal (emigration, transfer, and settlement) may be influenced by multiple factors operating across various spatial and temporal scales, which can affect dispersal syndromes or predictions of dispersal

ability[9,57]. We conducted the analysis at the species level and hence did not consider intra-specific variation in the dispersal syndrome or context dependence[55,58]. Future research should assess the consistency of dispersal syndromes at the population level[54,55,58–60] and integrate phenomenological kernels with mechanistic models that account for individual behavior, environmental context, and evolutionary processes[61].

Dispersal distance is a key parameter for process-based population models[11,62] that enable more robust predictions of movements, interactions, spatial genetic variation and species' ability to adapt to changing environments, and which ultimately allow for the development of effective strategies for conservation and management in the face of ongoing environmental challenges[11,63–65]. Our research underlines the existence and multi-faceted nature of dispersal syndromes, while showing that simple mechanistic traits

outperform complex syndromes in predicting dispersal ability. Altogether, our study calls for comprehensive, predictive and mechanistic frameworks to understand dispersal and its consequences under changing environments across diverse taxa and ecosystems.

## Methods

### Dispersal data

We gathered standardized estimates of median and long-distance dispersal from[8]. These are based on mark-recovery data from the EURING database[66] and from a methodological framework that addresses potential observation biases that provides total dispersal kernels (pooling all age classes) for 234 species, breeding dispersal kernel (between subsequent breeding attempts) for 113 species, and natal dispersal kernels (from natal site to first breeding site) for 122 species of European birds. We used the dispersal estimates from the Weibull and Half-Cauchy distributions because these clearly outperformed others for capturing rare long-distance dispersal events that contribute significantly to population dynamics[8]. In subsequent analyses, we used both the empirical median dispersal distance from the dispersal kernels and the long-distance dispersal measures, which were defined as the 95% percentile of the dispersal kernel. All source studies obtained appropriate ethical and regulatory approvals for the original data collection.

### Traits

Species' traits were represented by 7 variables related to morphology, life history, diet, behavior, habitat and range geography (Supplementary Table 1): (a) body mass, (b) Hand Wing Index, (c) diet niche, (d) life history strategy, (e) habitat preference in terms of landscape openness, (f) mean latitude of breeding range, and (g) migration distance between summer breeding and winter range. Body mass and Hand Wing Index (HWI), measured as Kipp's distance corrected for wing size, were extracted from ref. 16. Diet niche position was defined by an ordinal measure ranging from species feeding obligatory on plants (1) to species feeding obligatory on animals (5)[34]. Life history strategy was defined as the position of species along the slow-fast life history axis obtained by a principal component analysis of five life history traits extracted from ref. 35, namely egg mass, clutch size, age of first breeding, number of broods per season and life span. The first axis of the principal component analysis represents a gradient from long-lived species with high adult survival and low fecundity (slow pace-of-life) to early-maturing species with short generation times and high reproductive rates (fast pace-of-life). Habitat openness preference was defined as position of species' habitat niche along the gradient from forest interior (value = 1) to open treeless landscape (7)[67]. Finally, the mean latitude of breeding range was extracted from ref. 67 and the migration distance was measured as a great circle distance between centroids of species' breeding and non-breeding ranges[68]. Before subsequent analyses, all traits were scaled and standardized by subtracting the mean and dividing by the standard deviation[69] to have a mean of 0 and a variance of 1; body mass and HWI were log-transformed before scaling. Variance inflation factors (VIF) between all traits used in the analysis were always below 5. Because trait data were unavailable for all species in our data set, the number of species with the complete dataset was 138 species for total, 63 for breeding and 72 species for natal dispersal. We tried to develop a dataset with phylogenetic trait imputation to fill gaps in our data set. However, error estimations had high variability, and results were sensitive to the trait imputation (Supplementary Note 1), making it difficult to disentangle whether results differed because of the error estimation from trait imputation or because new species were included. Thus, for the subsequent analysis, we decided to proceed with only complete datasets (see Supplementary Note 1 for trait imputation results).

### Statistical analyses

#### Quantifying dispersal syndromes. 

We used Bayesian phylogenetic mixed models implemented in the package "brms"[70] to analyze the relationships between species traits and dispersal. We accounted for phylogenetic relationships among species by including a covariance

matrix containing phylogenetic distances among species. We obtained a species-level phylogeny on a sample of 500 trees obtained from the Hackett backbone of the global bird phylogeny (www.birdtree.org)[33]; trees were summarized into a single maximum clade credibility tree using the function *ls.consensus* from phytools[71] that computes the least-squares consensus tree from the mean patristic distance matrix of a set of trees. We computed branch lengths by the least-squares edge lengths computed on the mean patristic distance matrices from the phylogeny. The phylogenetic tree was transformed into a variance–covariance matrix[51] using the vcvPhylo() function in the package phytools[71]. We used a variance inflation factor analysis to account for potential multicollinearity, and all traits were retained in the model if VIF < 5. For each model, we obtained posterior distributions of all parameters by running 4 chains in parallel for 1000 iterations discarding the first 500 as burn-in. We used the function get_prior() in "brms" package to set uninformative, flat priors for the fixed effects[70]. Convergence was assessed by visually inspecting trace plots and ensuring that the R-hat parameter was close to 1 (≤1.02 in all cases). We report posterior means and their 95% credible intervals (CI) for all effects. Model r-squared values were computed using function r2_bayes() from the package "performance"[72]. Phylogenetic signal, or lambda (λ), was estimated from the models following the vignette and recommendations of P. Bürkner (https://cran.r-project.org/web/packages/brms/vignettes/brms_phylogenetics.html), using the 'hypothesis' method and substituting $\pi^{2/3}$ for the residual variance. Bayesian phylogenetic mixed models on dispersal distances (logged to meet the normality assumption) were fitted with a Gaussian distribution. We evaluated all models for overdispersion, normality, and multicollinearity using diagnostic functions in the performance package. Residual diagnostics were checked using simulations with the "DHARMa" package[73] and the functions from Dharma.helper[74].

We fitted models for each dispersal descriptor (median and long-distance dispersal), for each dispersal type (total, breeding and natal) and for the two main dispersal kernel distributions (Half-Cauchy and Weibull[8]). For each combination, we compared three model types: univariate models (single predictors), variable-selected models (optimized subsets), and complete models (all traits). We used the projection predictive inference for the selection of relevant predictor traits because it provides an excellent trade-off between model complexity and accuracy[75], especially when the goal is to identify a minimal subset of traits that yield a good predictive model. Model selection followed a systematic approach. First, we fitted complete models with all traits. We also tested ecologically meaningful trait interactions (body mass × life history, body mass × diet, body mass × habitat openness, and migration distance × latitude)[12,24], but we relegated these to Supplementary Note 4 to avoid overly complex models in the main analysis. Including these interactions produced similar results and did not alter the overall interpretation of the main effects. Second, we applied forward stepwise selection via projection predictive inference, adding traits that minimize Kullback-Leibler divergence[76] relative to the reference model. We selected optimal models based on expected log predictive density (ELPD) using 10-fold cross-validation, with the phylogenetic covariance term delayed to the final selection step to prevent it from capturing excessive variance. Third, we compared all model types using leave-one-out cross-validation (LOO) implemented in 'projpred' package[77] to rank model performance. When multiple models showed competitive performance (ΔLOO < 2 SE), we applied Bayesian model stacking to combine predictions and improve robustness. Final model selection prioritized parsimony while maintaining predictive performance (elpd_diff ≈ 0). The null model was never selected. (Supplementary Table 2). Because passerines are a much richer species group with less variation in morphological traits and other aspects of ecology and natural history, we ran a sensitivity analysis only with species from this Order (Supplementary Fig. 9). To ensure robustness of the variable selection approach, we also applied phylogenetic generalized least-squares models[78], applying model selection and averaging[76]. The consistency of key predictors across methods strengthens the reliability of the results (Supplementary Note 2).

### Evaluating and cross-predicting dispersal estimates

We tested the predictive performance of dispersal syndromes within and between bird orders, comparing our variable-selected multi-trait models and single-trait models. This way, we were able to ascertain how generalizable predictions of dispersal distances are across the bird phylogenetic tree when based on dispersal syndromes, including multiple traits or single traits. We selected four bird orders with a reasonable number of species to test within-order and between-order predictive performance (Accipitriformes 11 species, Anseriformes 12 species, Charadriiformes 11 species and Passeriformes 68 species) and used dispersal distances from the Weibull distribution[8].

Within-order predictive performance was assessed using five-fold cross-validation where species were partitioned into five folds, the multi-trait or single-trait models retrained on four folds and predicted to the hold-out fold of species[79]. Between-order predictive performance was assessed by training the multi-trait and single-trait models on each of the four bird orders and then predicting dispersal distances to the remaining orders. In all calibrated models, we included the covariance matrix containing phylogenetic distances among species. In the test, we allowed the predictions the possibility of including new levels on this covariance matrix, meaning that the prediction will use the unconditional values for data with previously unobserved levels. To examine the predictive power of the single-trait and multi-trait models, and the ability of models to predict dispersal distances between and within orders correctly, we used the function model_performance() from "performance" R package[72]. We used the r-squared value and the weighted mean r-squared value to evaluate the predictive performance[80], and ensure that the contributions of each group were proportional to the number of species (Supplementary Fig. 8).

### Statistics and reproducibility

All statistical analyses were performed in R version 4.4.1. Bayesian phylogenetic mixed models were fitted using the brms package with default priors and 4 chains of 1000 iterations each (500 warm-up). Convergence was assessed via R-hat values (≤1.02 in all cases) and visual inspection of trace plots. Model selection was performed using projection predictive variable selection implemented in the projpred package, with predictive performance evaluated via 10-fold cross-validation.

Sample sizes were determined by data availability: total dispersal analyses included $n = 138$ species, breeding dispersal $n = 63$ species, and natal dispersal $n = 72$ species. Cross-validation analyses were restricted to orders with ≥11 species (Accipitriformes $n = 11$, Anseriformes $n = 12$, Charadriiformes $n = 11$, Passeriformes $n = 68$). Each species represents an independent biological unit; no repeated measures or technical replicates were used. All analyses are fully reproducible using the code deposited on Zenodo[81].

### Reporting summary

Further information on research design is available in the Nature Portfolio Reporting Summary linked to this article.

## Data availability

All data used in this study are derived from previously published sources. Dispersal kernel estimates were obtained from Fandos et al.[8]. Species trait data were compiled from Sheard et al.[16], Reif et al.[34], and Storchová and Hořák[35]. Phylogenetic data were obtained from Jetz et al.[33]. No new data were generated in this study.

## Code availability

The code used to analyse the data and create the figures in this paper is available on Zenodo[81] under the identifier: Fandos, G., Robinson, R., & Zurell, D. (2026). Code and data for: Simple mechanistic traits outperform complex syndromes in predicting avian dispersal distances (all versions). Zenodo https://doi.org/10.5281/zenodo.10713957.

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

## Acknowledgements
D.Z. and G.F. received funding from the German Science Foundation DFG (grant no. ZU 361/1-1), G.F. also received funding from the Community of Madrid (Spain) and the Universidad Complutense de Madrid (Grant No. PR17/24-31914).

## Author contributions
G.F. and D.Z. designed research; G.F. performed research; G.F. analyzed data; G.F. and D.Z. conception and management of the project; G.F., R.R., and D.Z. wrote the paper. D.Z. acquired the funding.

## Funding

## Competing interests
The authors declare no competing interests.
