## [Transparent Peer Review file · Communications Biology]

Simple mechanistic traits outperform complex syndromes in predicting avian dispersal distances

Corresponding Author: Dr Guillermo Fandos

This manuscript has been previously reviewed at another journal that is not operating a transparent peer review scheme. The manuscript was considered suitable for publication without further review at Communications Biology.

Response #NCOMMS-25-30438-T

Dear Editor and Reviewers,

We sincerely thank both reviewers for their thorough and constructive evaluation of our manuscript. Their insightful comments have greatly improved the quality and clarity of our work.

Beyond addressing the specific reviewer comments, we have enhanced the comparison between single-trait and multi-trait models throughout the manuscript, improved the clarity of our main findings by focusing on the most robust predictors, and strengthened the ecological interpretation of our results. We have also conducted additional sensitivity analyses to support our conclusions.

We believe these revisions have substantially strengthened the manuscript and address all concerns raised by the reviewers. Below, we provide detailed responses to each comment.

REVIEWER #1

Comment 1: *"The correlational analyses yield some expected patterns, but the interpretation is relatively superficial and not well integrated with existing theory or previous empirical work. As a result, the discussion remains somewhat descriptive and misses an opportunity to advance broader understanding of dispersal ecology."*

Answer 1: We thank the reviewer for this helpful comment emphasising the need for stronger theoretical integration. We have thoroughly revised the manuscript following these suggestions. We have now substantially revised the Discussion to make these theoretical links more explicit and better highlight previous empirical work. We have also restructured the Results and Methods to focus on the main-effect models, moving interaction models to the Supplementary Information for clarity and transparency. These changes make the overall structure more coherent and improve the theoretical integration and interpretability of our results, while keeping the conclusions unchanged.

Comment 2: *"On the predictive side, while the authors outline a reasonable approach, they stop short of applying it to generate new estimates or demonstrate its utility in practice. This limits the potential contribution of the study."*

Answer 2: We generate predictions to new species through rigorous cross-validation analyses (Figure 3, Tables S2-S3). Our focus was on understanding the formation of dispersal syndromes and evaluating the predictive accuracy. Our results provide clear guidance for predicting dispersal ability for species lacking empirical data. Specifically, our results suggest predicting dispersal based on traits with clear mechanistic meaning such as body mass paired with phylogeny, especially for phylogenetically distant species.

While generating predictions for all European bird species would constitute a separate study requiring independent validation, we provide validated frameworks, quantified accuracy metrics, and clear implementation guidelines. This represents significant practical utility that enables future predictive applications while maintaining scientific rigor.

Comment 3: *"The use of interactions for some variables is not well justified. A couple of papers are cited justifying the inclusion of interactions body mass: habitat, and body mass: Life history. But other potential interactions were ignored, such as the potential interaction between HWI and migration, that was significant in Ref.19. A more detailed justification should be provided for why some interactions were included and others not."*

Answer 3: We thank the reviewer for this feedback. We have restructured our analysis following suggestions from both reviewers. The models were constrained by sample size and did not allow testing all potential interactions simultaneously. To be more transparent and improve interpretability, we have moved interaction models to supplementary material and now present main effects models as our primary results.

Our main analysis now focuses on variable selection among main effects and includes systematic comparisons with univariate model performance. This approach provides clearer interpretation, stronger predictive performance, and better addresses the reviewer's concern about interaction selection. The interaction models remain available

in supplementary material with detailed ecological justification for those specifically tested.

Comment 4: *"Figure 2. Fix this sentence: 'Lines correspond to univariate generalized linear mixed models accounting for phylogenetic relatedness phylogenetic generalized least-squares models.'"*

Answer 4: Thank you for catching this error. We have corrected the figure legend to read: "Lines correspond to univariate generalized linear mixed models accounting for phylogenetic relatedness" (Figure 2 legend).

REVIEWER #4

Comment 1: *"My only major concern is the four interaction terms, which I see that I share with a previous reviewer. It's not clear to me where these come from. I appreciate that the authors ran a version without interaction terms, but I think that either this version of the models should be the main version (and the interactions relegated to the supplement), or that the authors should more clearly justify why these interactions, but no others, were chosen."*

Answer 1: We thank the reviewer for this feedback and agree with this assessment. We have restructured our analysis accordingly, moving interaction models to supplementary material. The models were constrained by sample size and did not allow testing all potential interactions simultaneously.

To be more transparent and improve interpretability, we now present main effects models as our primary results, which are simpler to interpret and show stronger predictive performance. Our main analysis focuses on variable selection among main effects and includes systematic comparisons with univariate model performance. This approach provides clearer interpretation and better addresses concerns about interaction selection.

The interaction models remain available in supplementary material with detailed ecological justification for the specific interactions tested. This change has improved

both the clarity of our main findings and their practical applicability for predicting dispersal in species lacking empirical data.

Comment 2: *"L28-29: It's not especially clear, grammatically or scientifically, what 'or partially also by a proxy for flight efficiency' means. Consider rephrasing?"*

Answer 2: We have clarified this sentence to read: "...and to a lesser extent, flight efficiency" (line 28-29).

Comment 3: *"L47-48: The repetition here of the word 'thus' is a little clunky – consider rephrasing?"*

Answer 3: We have further revised this sentence for clarity and flow, as suggested. It now reads:

“Thus, a deeper understanding of dispersal traits and dispersal syndromes is also key to improving our predictive capacity and to evaluating the generality of trait-based approaches to anticipate future global change impacts on biodiversity”

Comment 4: *"L134: Something's gone amiss with the parentheses here."*

Answer 4: Thank you. We have corrected the parentheses formatting in line 134.

Comment 5: *"L147-149: I think it's sensible to check these models both across all birds and within the passerines... But I don't understand this particular assertion – perhaps consider rephrasing?"*

Answer 5: We thank the reviewer for this comment and have clarified the corresponding section to better explain our rationale. The revised text now reads:

“Overall, we found similar patterns and robust results when restricting the analysis to passerines (Fig. S3B.2), the largest avian order, which shares relatively similar life-history traits. This sensitivity analysis confirmed that the identified dispersal syndromes were consistent within a more phylogenetically and ecologically homogeneous group.” These revisions improve clarity and explicitly explain the purpose and outcome of the passerine-only analysis.

Comment 6: *"L283-285: This is a really important point, and possibly something that you might consider dwelling on slightly more. European dispersal (and migration) patterns simply might not generalise to other parts of the world!"*

Answer 6: We appreciate this observation and have expanded this discussion to emphasize the geographic limitations of our findings. We now explicitly acknowledge that "Distinct dispersal syndromes likely emerge in other biogeographic regions under different ecological constraints, seasonal patterns, and landscape structures "

Comment 7: *"L375: Not that it matters, but your font changes here."*

Answer 7: We have corrected the font formatting inconsistency.

Comment 8: *"L400: Typo, 'selected' rather than 'select'"*

Answer 8: Corrected to "selected" (line 400).

Comment 9: *"Figure 3: I found Figure 3 to be difficult to wrap my head around. In particular, I think I straight-up don't understand the conceptual illustration... Consider revising the top panel of this figure, and maybe adding a little more context for how the reader should be interpreting the lower panel?"*

Answer 9: We have revised the Figure 3 legend to clarify the cross-validation approach and make it more self-explanatory. The legend now explicitly explains the iterative nature of the between-order validation process and clearly defines what each visual element represents (point shapes indicating training orders, colors distinguishing within vs. between-order predictions, etc.). These clarifications should resolve the confusion about our methodological approach and make the figure more accessible to readers.

Comment 10: *"Supplement, page 5: The pronoun changes from 'we' to 'I' in one paragraph"*

Answer 10: We have corrected this inconsistency to maintain "we" throughout the manuscript.

Comment 11: *"Figure S3.2: Grammar in the last sentence of the legend: should be 'The species name for...is labeled' or 'The species names for...are labeled'."*

Answer 11: We have corrected the grammar in the supplementary figure legend.

Beyond addressing reviewer comments, we have enhanced the comparison between single-trait and multi-trait models throughout the manuscript and improved the clarity of our main findings by focusing on the most robust predictors. We have also strengthened the ecological interpretation of our results and conducted additional sensitivity analyses to support our conclusions.

We believe these revisions have significantly strengthened the manuscript and address all concerns raised by the reviewers. We hope the revised version meets the standards for publication and look forward to your feedback.

Sincerely,

The authors

Dear Editorial Team,

Thank you for the opportunity to revise our manuscript “Simple mechanistic traits outperform complex syndromes in predicting avian dispersal distances” (COMMSBIO-25-10669A). We appreciate the constructive feedback and have addressed all concerns. Below, we summarise the main changes.

Reviewer #1, Comment 2 (Limitations and scope) We have expanded the Discussion to explicitly acknowledge that our analyses are restricted to European birds and that trait-based predictions require further downstream validation. The statement regarding “the most comprehensive study” has been qualified with more conservative language.

Reviewer #1, Comment 3 & Reviewer #2, Comment 4 (Interaction models) Following both reviewers’ suggestions, main-effect models are now presented as the primary analysis. Interaction models have been moved to Supplementary Note 3, where we provide clear biological justification for each interaction term (body mass × life history, body mass × diet, body mass × habitat openness, migration × latitude).

Reviewer #1, Comment 4 (Figure 2 typo) Corrected.

Additional editorial changes - Sample sizes (n) and error bar definitions added to all figure legends - Statistics and Reproducibility section added to Methods - Supplementary Information restructured with sequential numbering (Figures 1–16, Tables 1–8, Notes 1–4) - “K-selected”/“r-selected” terminology replaced with descriptive explanations - Data Availability and Code Availability statements included, with code deposited in Zenodo (<https://doi.org/10.5281/zenodo.17508963>)

A point-by-point response to all editorial comments is on the Editorial Requests Table.

We believe the revised manuscript fully addresses the reviewers’ concerns and hope it now meets the standards for publication in *Communications Biology*.

Yours sincerely,

Guillermo Fandos On behalf of all co-authors